# Implications of Extractivism and Environmental Pollution in Mapuche Territories of the Araucania Region

**DOI:** 10.3390/ijerph20095672

**Published:** 2023-04-28

**Authors:** Juan Beltrán-Véliz, José Luis Gálvez-Nieto, Julio Tereucán-Angulo, Fabián Muñoz-Vidal, Nathaly Vera-Gajardo, Pablo Müller-Ferrés

**Affiliations:** 1Núcleo Científico Tecnológico en Ciencias Sociales y Humanidades, Universidad de La Frontera, Temuco 4811230, Chile; 2Departamento de Trabajo Social, Universidad de La Frontera, Temuco 4780000, Chile; 3Departamento de Educación, Universidad de La Frontera, Temuco 4780000, Chile; 4Facultad de Educación, Universidad Autónoma de Chile, Temuco 3480094, Chile; 5Facultad de Administración y Negocios, Universidad Autónoma de Chile, Temuco 7500912, Chile

**Keywords:** extractivism, consumerism, environmental crisis, küme mogen, environmental education

## Abstract

Chile is facing an environmental crisis and the territory of the Mapuche people is no exception. This is largely due to extractivism, which refers to the massive extraction and exploitation of natural resources in an indiscriminate manner. The objective of this study was to reveal the implications of extractivism and environmental pollution in Mapuche territories in the Araucanía region. The methodology used was qualitative, based on constructivist grounded theory. In-depth interviews and participant observation were used to collect data. The participants were 46 kimeltuchefes. The main results revealed extensive monocultures of non-native trees: pine and eucalyptus, which consume large amounts of water. They also revealed environmental pollution and indiscriminate forestry extractivism related to these trees, which generate soil degradation and water pollution. These consequences reduce biodiversity and disturb the ngenh (spiritual beings and protectors of nature). They also affect the Mapuche’s agricultural activities and, in turn, their health and subsistence. In addition, non-native tree monocultures, environmental pollution and forestry extractivism transgress the az mapu (Mapuche code of ethics and behaviour), which disturbs the ethical, moral and spiritual relationship between the Mapuche and nature. They also have negative implications for the küme mogen (good living of the Mapuche), since they violate the balance and harmony between the Mapuche and all living beings, elements and spiritual beings that are part of nature. This also violates the reciprocity between the Mapuche and nature. It was concluded that there have been violations of the human rights of the Mapuche people, given that they are exposed to harmful environmental conditions that put their health and subsistence at considerable risk. In this sense, the Mapuche are experiencing a spiritual, physical, cognitive, attitudinal, affective and material imbalance. Ultimately, the state of Chile must generate intercultural environmental public and educational policies aimed at generating environmental awareness and creating actions to solve environmental problems in order to protect Mapuche and non-Mapuche territories.

## 1. Introduction

One of the main problems and global priorities identified by the WHO for 2019–2023 that humanity currently faces is climate change [1], as planet Earth is undergoing indiscriminate environmental pollution [2], resulting in unsafe water and causing the deaths of 1.7 million children under five years of age [3]. This is largely due to the extractivist and mercantile economic model operating in most regions of the world, which does not consider pollution and the degradation of natural systems. This has led to substantial deterioration of the environment due to water scarcity and soil degradation, which, in turn, causes loss of biodiversity [4,5,6]. It also greatly affects human health [7,8]. In this regard, a study carried out in the Loreto region of Peru showed that 82.7% of the population studied affirmed that extractivist activity had affected the water and food they consume. Additionally, 82.3% stated that they had become ill, which manifests itself as vomiting, diarrhoea, stomach pain, headache, infections, fever, skin problems, parasitosis and bronchitis, due to the consumption of water and contaminated fish [9]. In this sense, human beings and other forms of life are experiencing health problems and facing a shortage of natural resources and food, which translate into poverty, hunger and death [10,11]. Extractivism and environmental pollution in the Chilean context have caused a major environmental crisis [12,13,14] since pollution is permanently high; the habitat is being lost, biological resources are becoming depleted and water is scarce and polluted. Thus, the population is very vulnerable to climate change, making quality of life deteriorate [15,16].

In this context, the extraction of natural resources is facilitated by a legal regime [17]. In the territory of the Mapuche people, this type of activity is no exception due to the public policies fostered and protected by the Chilean state; particularly, Law 21.162, which is related to the extractivist model [18,19]. Large national and transnational companies act in alliance with the state in the territories of the Mapuche people [20,21,22] to extract natural resources. This is configured as a structural lock against any modification of the environmental policy [18]. Thus, these forestry companies have transgressed and continue to transgress the spaces that constitute the Mapuche territories since they have replaced the native forest with eucalyptus and pine monocultures [23,24] in order to exploit and commercialize them. Likewise, large quantities of pesticides are used, causing soil deterioration and contamination of different areas and water bodies; at the same time, excessive water consumption can be observed, affecting all forms of life that live there [24,25,26]. Therefore, this type of activity results in bad environmental conditions in the Mapuche territories, which, in turn, generate vulnerability and have negative impacts on the health of the members of the Mapuche communities [27,28]. It is difficult for these activities to coexist with the other productive activities that are typical of the area, thus affecting the subsistence of the Mapuche people [29] and disturbing the way that Mapuche individuals relate to nature and, therefore, their ways of being and living [22,30]. In view of the previous information and the problems described, this study aimed to reveal the implications of extractivism and environmental pollution in Mapuche territories in the Araucanía region.

## 2. Theoretical Framework

### Extractivist Economic Model and Environmental Pollution in the Mapuche Context

The Chilean economic model is a neoliberal, extractivist model based on the progressive commercialization of nature [31]. The extractivist model is based on the appropriation, extraction, exploitation and massive accumulation of natural resources on a large scale, and their export to international markets employs an inadequate process [17,32,33]. In this way, all valuable natural resources are appropriated, exploited, quantified and utilized to expand and sustain the neoliberal economic model [23,34,35]. This generates capitalization in the form of money, which, in turn, leads to profits and increased consumption that sustain and promote the extractivist economic system [36,37]. Resource extractivism in Chile depends on the existence of large amounts of land and water, which leads to soil deterioration and water scarcity [24,38].

The forestry industry in Chile represents about 2% of the GDP and more than 8% of total exports, making it the third largest economic sector in the country [39]. In this context, the accumulated area of forest plantations in Chile as of December 2018 (latest available figure) amounts to a total of 2.303.886 hectares. Radiata pine plantations cover most of this area with 55.8% of the total, followed by eucalyptus plantations with 37.2% [40]. Forestry activity is mainly present in Mapuche territories in the form of monoculture plantations, especially pine and eucalyptus, which result in high water consumption and contamination of water at the ground and surface levels, as well as soil degradation; therefore, such activities generate environmental pollution [24,26,41,42,43]. This affects the daily life activities and subsistence of the Mapuche people, altering their way of life, as well as all forms of life that inhabit and coexist with nature [10,44]. In fact, forestry companies operating under the consent of the Chilean state in Mapuche territories generate environmental damage and pollution and, therefore, demonstrate a lack of respect and care towards nature and the worldview, traditions, beliefs and ways of life of the Mapuche people, which are closely connected to nature.

## 3. Materials and Methods

### 3.1. Participants

A qualitative methodology was applied to capture reality in the context in which the phenomena occur, employing interpretation and detailed description based on the meanings of the experiences of the participants, which underlie the socially constructed reality used for the understanding of sociocultural phenomena [45,46]. From this perspective, ethnography can be used to study the culture of groups—in this case, that of the kimeltuchefes—in order to describe, analyse and interpret it as a whole. This, in turn, makes it possible to understand the behaviour of the subject from a natural perspective and their relationship with the environment based on their actions and meanings [47,48].

On the other hand, constructivist grounded theory was employed, and the narratives and social constructions of culturally situated subjects were used as sources for the construction of theoretical categories [49]. It was in this sense that the narratives of the kimeltuchefe people located in the Mapuche communities of the Araucanía region were considered. Observations were also considered in this context.

The selected participants were 46 kimeltuchefe people located in rural Mapuche communities. They had the Mapuche kimün—Mapuche knowledge in the different fields of human development: social, cultural and economic, among others. The kimeltuchefe are Mapuche people who carry knowledge, wisdom and experience. In turn, they teach Mapuche knowledge and educate and socialize Mapuche people. Among the participants were: (a) Logko (a leader who directs the decisions of the community to which he belongs); (b) Kimche (a person who has the social and cultural knowledge of Mapuche society); (c) Lawentuchefe (he or she recognizes and uses medicinal plants and practices the healing methods that are applied by people); (d) Norche (a person who always acts in righteousness and who is naturally based on the wisdom of things); and (e) Ngenpin (an authority who directs the guillatún (rogation days)). These participants were located in the Pewenche (people of the Pewen), Lafkenche (people of the sea) and Wenteche (people of the valleys and riverbank of the Cautín River) territories. Below, in Table 1, the summary of the sample is presented.

The participants were selected intentionally, which made it possible to select those with the best knowledge of the phenomena studied. This guaranteed an effective and efficient saturation of the categories and subcategories [50]. In this context, the following inclusion criteria were used for the selection of participants. Table 2 below shows the inclusion criteria of the participants.

It should be clarified that the inclusion criterion referring to people over 60 years of age was legitimate because kimeltuchefes are people who have acquired the ability and methodology to teach during their lives. They are capable of delivering the Mapuche kimün [51]. Likewise, De Augusta [52] points out that, in the case of Mapuche culture, the bearers of knowledge are the grandparents, who belong to the category kimeltuchefe and, in turn, are responsible for educating and transmitting the Mapuche kimün to the new generations.

In order to select the participants, an intercultural facilitator was required who had the Mapuche kimün. For this purpose, the researchers, together with the intercultural facilitator, made visits to different Mapuche communities in order to find people who belonged to the category kimeltuchefe.

### 3.2. Instruments

To achieve the intended objective, the following data collection instruments were applied: in-depth interviews and participant observation. Regarding the in-depth interviews [53], they did not follow a pre-established and standardized scheme but had a flexible form in their structuring so that the subject of study could share the meanings required by the interviewer to better understand the phenomenon being studied. The type of questions that were asked to the kimeltuchefes were open and in-depth, aiming to capture all the richness of their meanings.

The participant observation ranged from active intervention to simply being present in the studied scenarios [54]. In this study, the researchers were present in the spaces of the Mapuche territories, observing practices, situations and the events that occurred.

### 3.3. Procedure

The interview technique was verified by means of a pilot test. In this regard, the authors of [55] describe the purpose of such tests as simulating the implementation of the interview, adjusting the format and establishing characteristics referring to the moment of application, as well as, therefore, visualizing the effectiveness beforehand through simulated practice. The test of the interview (piloting) made it possible to adjust the script for the questions and delve deeper into the phenomenon under study. The piloting was conducted with three kimeltuchefe people with similar characteristics to the participants of this study.

Subsequently, contact was made with the logkos, who were given a formal letter requesting authorization from their community to conduct the study and access information from the participants, as well as indicating the purposes of the study and its ethical aspects. Once the authorization was provided by the logkos of each community, the kimeltuchefe people were given a consent form with which they could voluntarily give their consent to participate in the research when it agreed with their values, principles and the interest they had in describing their experience of the studied phenomenon, as long as this participation did not result in detriment of any kind [56]. The identities of the subjects and the information provided by them were protected. In addition, they were informed that, once the study was concluded, the findings would be sent to them. Finally, it should be noted that the consent forms were previously approved by the Ethics Committee of the Universidad de La Frontera. Once the consent forms had been signed by the kimeltuchefe people, the instruments were applied.

### 3.4. Data Analysis

The information collected through in-depth interviews and participant observation was reduced and analysed with Atlas ti 7.0 software (Susanne Friese, Göttingen, Germany). Constructivist grounded theory [49] was used to analyse the data and create categories and theoretical subcategories from the data. In the data analysis process, open coding was used in the first stage and axial coding in the second stage. In the open-coding phase, the text was accessed and the ideas, thoughts and meanings contained in it were extracted in order to discover, name and develop the concepts [57]. The data were disaggregated into discrete parts, scrutinized and compared for differences and similarities in order to create categories and subcategories [57]. Subsequently, in the axial coding, the categories were associated with their subcategories [57].

During the analysis process, “memos” were used that permanently supported the researchers in expressing thoughts, asking questions and producing reflective notes and analyses, helping to identify themes and patterns in the data and contributing to increasing the level of conceptualization of the phenomena under study [58,59]. This procedure was performed when the information emanating from the subjects was confusing and insubstantial, requiring the researchers to return to the research field in order to re-interview the kimeltuchefe people and capture in depth the richness of their meanings, thus clarifying the information. This helped enrich the categories, subcategories and the relations between them in a more precise way.

In the coding process, the categories “forest extractivism and environmental pollution” were developed, which were related to the subcategories “Monocultures of non-native trees”, “Environmental pollution”, “Forest extractivism”, “Az mapu” and “Küme mogen”. These subcategories were carefully quantified (see Appendix A) based on the participant interviews, allowing the identification of numerical patterns that facilitated the identification and saturation of the subcategories [58,60].

The entire analysis process was conducted within the framework of a constant comparative method (CCM). The researchers carefully compared the codes that emerged from the data in a systematic way with the codes and classifications obtained [60]. In addition, the CCM enabled the triangulation of the information to avoid biases and deviations. Furthermore, in a complementary manner, confirmability was used to analyse and interpret the data from a neutral perspective [50]. This was carried out by the researchers and the intercultural facilitator who participated in the study and who, in addition, made continuous returns to the homes of the kimeltuchefe people—where spaces for conversation were established—in order to verify the findings as real and true [50].

## 4. Results

The network shown in Figure 1 resulted from the in-depth interviews and participant observations through a process of open and axial coding using interpretation. The subcategories presented here are those that presented the highest saturation.

The results derived from “forest extractivism and environmental pollution” category are presented, which were related to the following subcategories: (a) “Monocultures of non-native trees”, (b) “Environmental pollution”, (c) “Forest extractivism”, (d) “Az mapu” and (e) “Küme mogen”. This can be seen in the following Figure 1.

### 4.1. Category 1: Forest Extractivism and Environmental Pollution

In this category, the negative implications of forest extractivism and environmental pollution in Mapuche territories were visualized. At the same time, it was observed that the Mapuche way of life and all forms of life that inhabit nature are negatively altered.

#### 4.1.1. Subcategory a. Monocultures of Non-Native Trees

There are low levels of water flow in the rivers, particularly in the river that flows through the Rulo sector; this is due to the “monoculture of non-native tree plantations”, such as pine and eucalyptus: “Near the rivers there are large pine and eucalyptus plantations, here in Rulo, it is possible to observe that there is a low water level, this is largely due to the excessive monoculture planting of these non-native plantations” [kimche]. In this regard, “large extensions of this type of plantations are observed near the banks of the Rulo River and other rivers, where low water flows are visible” (observation). This can be seen in the following Figure 2 and Figure 3.

In addition, these plantations have affected sacred spaces, such as menokos (wetlands with great biodiversity and an abundance of medicinal herbs where the lawen is found (Mapuche ancestral medicine)), since they consume large quantities of water, causing water shortages in the menokos. This has implications regarding the decrease in biodiversity, as reflected by the lack of medicinal herbs, which, in turn, negatively affects the health of the Mapuche people. This is indicated in the following narrative:
The menokos have been drying up. Our community is surrounded by 3 estates with these plantations, so we have no lawen, we have lost medicinal plants, and we have to go far away to search for them.[Lawentuchefe]

#### 4.1.2. Subcategory b. Environmental Pollution

“Environmental pollution” was observed due to the use of pesticides in the pine and eucalyptus plantations. These pesticides cause considerable contamination of soil and water and, at the same time, affect the subsistence of the living beings that inhabit this territory and their ways of life. They also disturb the spiritual beings who are part of and protect these spaces. This is evidenced by the following narration: “The land is sick because of the pine and eucalyptus plantations, ahh…, and also because of the use of pesticides on these trees, they contaminate everything, since they affect the water, soils, living beings and the Ngenh (spiritual beings) of those spaces” (Kimche). In this context, other interviewees added that these pesticides affect the health of the Mapuche people who live there as they consume contaminated water and contaminated food that comes from crops or grows naturally on these lands. This has repeatedly caused intoxication, vomiting, diarrhoea and headaches. This is visualized in the following narrations:
The forestry companies use chemicals, how could I say it…? They apply pesticides to the pine plantations, that damages the soil, contaminates the water of the rivers, lakes, etc., and the other plants…, better said, they damage all the living beings that live in those places. We drink contaminated water, and we eat contaminated food that we grow in these areas or that come from these areas, this affects our health.[Logko]
These people who have pine and eucalyptus plantations fumigate them with pesticides that are very toxic to our health, they don’t care. I got sick, I had severe headaches, vomiting, diarrhea. Also, my peñis (brothers) and lagmienes (sisters) have become sick due to this contamination, because the soil and groundwater are contaminated with pesticides. So, you get sick when you consume wild foods that grow in these places, haaa… and also, from the food that you grow here to eat.[Ngenpin]
These soils have many chemicals, the waters that are under the earth are contaminated by these chemicals that the forestry companies apply to these eucalyptus plantations. They do great damage to us. The water from the well is contaminated, I drink from that water, and I have been intoxicated, I was very sick. This has happened to me several times, because the water is contaminated, and the only way to get water here is through the well.[Norche]

#### 4.1.3. Subcategory c. Forest extractivism

In this subcategory, indiscriminate “forest extractivism” in the form of non-native tree plantations, such as pine and eucalyptus, was described, which erodes and degrades the land and causes water shortages. This has implications in the form of considerable losses in biodiversity, and it also negatively affects the agricultural activities performed by the Mapuche people, which, in turn, results in hunger and poverty among the members of Mapuche communities.
When one walks through these large areas of land where these non-native tree species have been cut down (…) one sees the land with a reddish color, the soil is weathered, eroded, there is a lack of water. There is almost no life on these lands, due to extractivism. Ahh…, this also affects the planting of wheat, legumes, potatoes, vegetables, etc., which affects the Mapuche communities, because one sees poverty and hunger.[Kimche]

Extractivism in the form of non-native plantations can be observed systematically, and these plantations are later commercialized and capitalized in the form of money, which generates large profits for forestry businessmen. “It is possible to see large quantities of pine and eucalyptus plantations, which are not native to Mapuche territory, and which are exported and capitalized in cash” (observation). This corresponds with the following narrative: “They cut these plantations and then sell the wood… they make a lot of money. This is how they do it over and over again, this damages the land a lot” (Logko). This can be seen in the following Figure 4.

#### 4.1.4. Subcategory d. Az mapu

The Mapuche people are guided by the az mapu, which refers to an ethical and moral code that regulates the behaviour of the Mapuche people in relation to nature. In this context, respect and care for the spaces and for all the elements and beings that conform to nature are observed. For the Mapuche people, all elements and living and spiritual beings that are part of nature have life and, consequently, value and spiritual meaning. The Ngenh have special relevance because spiritual beings protect the spaces of nature and other forms of life. However, for the Wigkas (non-Mapuche people), the rocks, hills and volcanoes and the Ngenh of these spaces lack value and ethical and spiritual meaning. This implies a limited view on the part of non-Mapuche people with respect to valuing and relating to nature. This can be seen in the following narratives.
We, the Mapuche people, respect and take care of all nature, we are guided by the az mapu, to maintain a relation of respect with nature. For the Mapuche people, a rock has life, for you Wigkas, it has no life, for us that rock has life and a meaning, because there is a Ngenh that protects that rock. Example: if we look at a rock, volcano, or a hill, for you Wigkas, it has no life, no meaning, but for us it has life and meaning. Also, in the rocks there are small living beings, in the hills there are plants, herbs, animals and insects, etc.[Norche]
The Winkas see the land as inferior, they do not have any respect for it. They come and grow big plantations; they don’t care about the lives that are in those spaces. For us every space, life and element are important, because for us everything has life, the hills, forests, water, stones, birds, insects, etc. All that is protected by a Ngenh.[Ngenpin]

#### 4.1.5. Subcategory e. Küme mogen

The purchase of large amounts of land by national or transnational forestry companies for the planting of non-native trees was observed. This type of activity affects the soils and the diversity of life and alters the relationship and balance between the Mapuche people and nature, as well as, therefore, the küme mogen of Mapuche individuals (good living). For this reason, the Mapuche people maintain a firm attitude towards the protection of nature:
A national or transnational company comes, buys an estate, buys an average of 200 to 400 hectares, and then plants them with pine or eucalyptus. The land gets sick with these species, affecting all the beings that live there. The küme mogen cycle is broken, because the monoculture of these species does not allow the existence of other living beings. So, the balance between the Mapuche individual and nature is broken, that is why the Mapuche society defends nature so much.[Kimche]

This type of plantation negatively intervenes in the relationships and balance between all living beings and elements that are part of the sacred spaces of nature; that is to say, it disturbs the “ixofil mogen” (all types of life, without exception) and the bonds of harmony between the Mapuche people and nature, causing an imbalance in the Mapuche individual from the physical, cognitive, moral, material, spiritual and affective points of view. In order to avoid this, a llellipun should be performed to establish a spiritual connection between the Mapuche individual and the Ngenh, employing newen (energy) to makes requests of and thank the Ngenh of that space and, in this way, maintain a state of balance and well-being with nature:
The transgression of forestry companies in the sacred spaces of the Mapuche territory with eucalyptus and pine plantations disrupts the relationships and balance of all living beings and elements that are in those spaces, for example: birds, insects, plants, herbs, water, stones, rocks, spiritual beings, the way I could explain it is that it transgresses the ixofil mogen, this represents for us that everything has life. Also, this directly affects the relationship of balance between the Mapuche people and nature. We, Mapuche individuals, get sick, we become physically, spiritually, and materially unbalanced, and the emotional part is also affected. To avoid this, we respect nature, so we must ask for permission from the Ngenh of that space. For example, to go to a wingkul (hill), trayenko (waterfalls), menoko, lemu (forest) etc., or to drink water from a river, eat some fruit from a tree or plant, we ask for permission and thank the Ngenh of those spaces. We do this with a llellipun (ceremony), in this invocation spirituality is present, ahh…, this is related to the newen. So that allows you to be in balance and have a good living.[Ngenpin]

On the other hand, the following story depicts the principle of reciprocity, which is manifested in the mutual help shared by Mapuche individuals and nature. In this context, nature provides the Mapuche people with food, knowledge and protection. At the same time, the Mapuche people show respect, care and love for nature and, in turn, they give it grains, seeds and silver coins. This is related to being Che (person), which is especially relevant in Mapuche culture, since it is based on the moral and spiritual dimension:
Nature is like a mother to us, because she takes care of us, and we respect and protect her, so there is mutual help. Nature gives us food, she shelters us ahh… and gives us a lot of kimün. We thank nature for what she gives us, and we give her seeds, grains of what was harvested, muday, and also old silver coins. This is related to being Che, which for us is very important, we always teach to be Che, since a person must be clean in thought, heart, deed, and spirit.[Norche]

## 5. Discussion

The “monocultures of non-native trees” have implications for the Mapuche communities, as can be seen in the low water levels in the rivers. Furthermore, they generate dryness in sacred spaces, such as the menoko, since these plantations consume large amounts of water, causing water shortages [29,64]. This has implications in terms of the reduction in the biodiversity [23,65] that exists in the area. There is a reduction in the availability of medicinal herbs in the menoko, which negatively affects the health of the Mapuche people. Correspondingly, studies have shown a decrease in the range of herbs and medicinal plants due to extractivism and environmental pollution [21,66,67], and these plants are in danger of extinction and losing their ancestral medicinal use value [68]. For this reason, the Ministry of Agriculture recovered native plants of ancestral Mapuche medicinal use through a project, giving them value [68]. Likewise, the Universidad de La Frontera worked on the recovery of medicinal plants in danger of extinction through a project, including salvia, huella chica, pichi romero, menta de árbol, palo negro, copihue, boldo and canelo [69].

This type of forestry activity also generates “environmental pollution” due to the use of pesticides in the plantations, which considerably degrade and contaminate soils and water and, therefore, affect biodiversity and the health of the Mapuche people [24,26,70]. In relation to the above, a systematic review involving several regions of Chile, including the Mapuche territories, demonstrated exposure to pesticides and their effects on the health of the population. The most widely observed effects in this population were neurotoxic (54%), genotoxic (31%) and reproductive (15%) effects. In Chile, the levels of exposure among Mapuche and non-Mapuche populations to pesticides are higher than those found in international studies [71]. Furthermore, research shows that pesticide spraying where there are streams and wetlands has disturbed the growth of herbs and a variety of medicinal plants [66]. This is evidenced by the fact that most of the Mapuche have become ill, as they suffer from intoxication, vomiting, headaches and diarrhea. The latter is due to the consumption of contaminated water and contaminated food that is grown on or comes directly from these lands. Pesticides are potentially toxic to humans and can have intense and chronic effects on people’s health. In addition, pesticide residues are found in both animal and vegetable food and water, which generates a considerable risk for the health of consumers [72].

Likewise, the disturbance of the Ngenh who are part of and protect these spaces [73]—and who are also connected to the spirituality of the Mapuche people—is also visible.

The indiscriminate “forest extractivism” related to non-native tree species causes water scarcity, and it also generates erosion and leads to loss of soil nutrients [24]. This severely affects biodiversity [74,75], which has negative implications for the development of the agricultural activities performed by the Mapuche people, affecting the subsistence of their communities [21,29] and resulting in hunger and poverty. Furthermore, extractivist activity is based on the continuous exploitation of the natural resources [76] represented by these non-native species, which are commercialized and capitalized in the form of money [37,77]. In this way, the extractivist economic model [34,78] is sustained and stimulated over time.

In this context, “monocultures of non-native trees”, “environmental pollution” and “forest extractivism” transgress Mapuche spaces and, therefore, the az mapu, which corresponds to a code of ethics and behaviour for the Mapuche people in relation to nature, the surroundings and the environment [79] based on a variety of rules, procedures and protocols [80]. This code regulates and orients the Mapuche people with regard to their way of relating to nature, given that all elements and living beings that are part of nature, as well as the Ngenh, have life. In this context, the Ngenh become vitally important, given that they are spiritual beings; i.e., superior beings and owners and protectors of spaces in nature [73,81] and of every element and living being. This helps in understanding that Mapuche individuals are part of nature and, therefore, they maintain a horizontal and harmonious relationship with it. In this sense, the Mapuche people maintain a relationship of respect and care with nature, and this has a moral [10], ethical and spiritual meaning for the Mapuche.

In contrast, for the Wigka people, hills, rocks, volcanoes and water, among other components, elements and spaces, lack life and, therefore, have neither meaning nor value. Indeed, non-Mapuche people have a limited and reductionist way of perceiving, valuing and relating to nature. This is largely due to the form of scientific explanation they use for reality, which is based on the dominant positivist rationality [50] that has been and continues to be present in different areas of knowledge [82]. In this sense, one kind of knowledge is possible: Western scientific knowledge [83].

Forestry activities involving plantations of non-native species, which result in forestry extractivism and environmental pollution, as well as affecting soils and biodiversity, alter the relationship and balance between Mapuche individuals and nature and, consequently, the küme mogen of the Mapuche people, which allows them to maintain a harmonious relationship with themselves, with others and with nature [84]. Furthermore, these forestry activities disrupt the connections and balance of the ixofil mogen: all types of life forms, without exception, that interact with each other and are interdependent [85]. This type of violation has implications for the Mapuche way of life, given that, in contrast, the Mapuche people maintain an attitude of respect for all spaces, living beings, elements and spirits that integrate with and coexist in nature. In fact, the Mapuche people maintain a solid attitude towards the protection of nature, since they understand that the land for them is something paramount from which they cannot isolate themselves [86].

In this context, breaking the harmony between the Mapuche people and nature causes an imbalance in Mapuche individuals from the physical, cognitive, moral, material, emotional and spiritual points of view. To avoid this imbalance, a llellipun must be performed, which is a ceremony that translates into a prayer to establish a spiritual connection between the Mapuche people and the Ngenh. In this ceremony, newen is used to request entrance to a space from the Ngenh of that space and, at the same time, to thank this spirit for what the space will provide. In this way, a state of balance and well-being with nature is maintained [81]. It should be noted that, in order to establish the connection between the Mapuche people and the Ngenh, there must be newen, which corresponds to the spiritual energy [87] that is generated between these two beings. This energy does not originate by itself but in connection with the elements—water, fire, earth and air—and the kimün—the knowledge, discernment, thought, values, attitudes and tools necessary to communicate with everything that exists in nature [88].

This is closely related to the principle of reciprocity, which refers to a commitment and a responsibility maintained through a relationship between the Mapuche people and nature based on mutual help [89] and oriented towards the greater good [90]. This is reflected in the fact that nature grants food, protection and knowledge to the Mapuche people [10,91]. In turn, the Mapuche people show respect, love and care for nature [19] and also give it silver coins, muday, grains and seeds. All this is connected with being Che, which is of vital importance in Mapuche culture, since it is based on ethical, moral and spiritual aspects. In this regard [92], researchers have indicated that the Che category is the ethical and moral axis leading to the formation of the Mapuche individual that allows him/her to be in harmony with nature [88]. In this sense, it should be noted that the process of becoming Che is constantly in movement, allowing the Mapuche to develop and behave appropriately and integrally with the environment, nature and a diverse society.

## 6. Conclusions

It was found that large forestry plantations employing “monocultures of non-native tree species” consume large amounts of water, which reduces river flows and alters the biodiversity of sacred spaces, including the menoko, which, in turn, negatively affects the health of the Mapuche people.

The “environmental pollution” from the use of pesticides and the indiscriminate “forest extractivism” related to these tree species generate degradation and contamination of soils and water. Therefore, they severely disturb all types of life that exist in the area, including the Ngenh. This also affects the agricultural activities performed by the Mapuche people and, in turn, their health, as reflected by intoxication, vomiting, headaches and diarrhea. It also affects the survival of their communities. In addition, the systematic extraction of these forest resources is oriented towards commercialization and capitalization in the form of money, which helps to sustain and foster the extractivist economic model even more.

In addition, it was noted that “monocultures of non-native trees”, “environmental pollution” and “forest extractivism” transgress Mapuche spaces and, consequently, the az mapu, which affects the ethical and behavioural relationship between the Mapuche people and nature since, for Mapuche individuals, nature has value and ethical and spiritual meaning. However, non-Mapuche people have a reductionist view of how to value and relate to nature, which is related to the explanation of reality using positivist rationality. This has negative implications for the küme mogen (good living) of the Mapuche people, since it violates their balance and harmony with all living beings, spiritual beings and elements that are part of nature. At the same time, the bonds and balance of the ixofil mogen are disturbed. This causes an imbalance in Mapuche individuals from the physical, cognitive, attitudinal, affective, material and spiritual points of view. In order to avoid such transgression and imbalance, a llelipun should be performed to request entrance to a space from the Ngenh and, in turn, to thank the Ngenh for what the space will offer.

The aforementioned is closely related to the reciprocity between the Mapuche people and nature. This reciprocity is reflected in the respect, love and care Mapuche individuals have for nature, which in turn provides food, knowledge and protection to the Mapuche people. In addition, reciprocity is related to the condition of being Che (person), which is based on and constituted by the spiritual, ethical and moral; at the same time, it is in constant construction. This allows each Mapuche individual to maintain a harmonious relationship with himself/herself, with nature, with his or her environment and with a diverse society. Consequently, it allows him or her to achieve the good life.

In summary, it is possible to confirm violations of the human rights of the Mapuche people, given that they are permanently exposed to harmful environmental conditions that put their health at considerable risk due to the contamination of soil, water and food. This also generates transgressions in the Mapuche’s ways of relating to nature according to their worldview and, therefore, in their ways of perceiving, thinking, feeling, acting and living.

In view of the conclusions, it is necessary to move from an extractivist economic model to a sustainable and renewable economic model. In this sense, the state of Chile should generate environmental public policies aimed at protecting Mapuche and non-Mapuche territories from the environmental crisis and develop educational public policies with an intercultural environmental approach in the school and higher education contexts. This will contribute to the recognition, understanding and acceptance of the other, considering their differences, and make it possible to generate environmental awareness, take measures and create actions to solve the problems that underlie the environmental crisis based on dialogue and constant collaboration. In addition, it will contribute to the integral development of Mapuche and non-Mapuche people and help them achieve the good life.

This study was part of the DIUFRO investigation: DFP19-0049.

## Figures and Tables

**Figure 1 ijerph-20-05672-f001:**
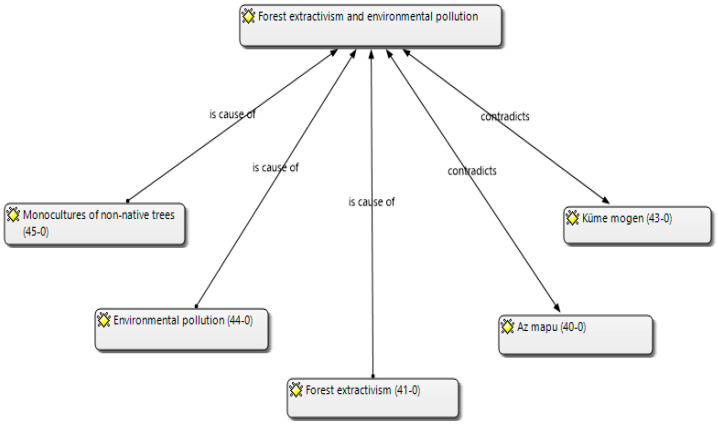
Forest extractivism and environmental pollution.

**Figure 2 ijerph-20-05672-f002:**
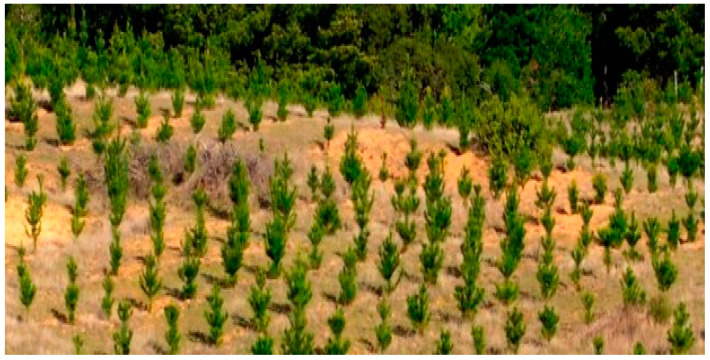
Pine plantations in Mapuche territories (July 2021) [61].

**Figure 3 ijerph-20-05672-f003:**
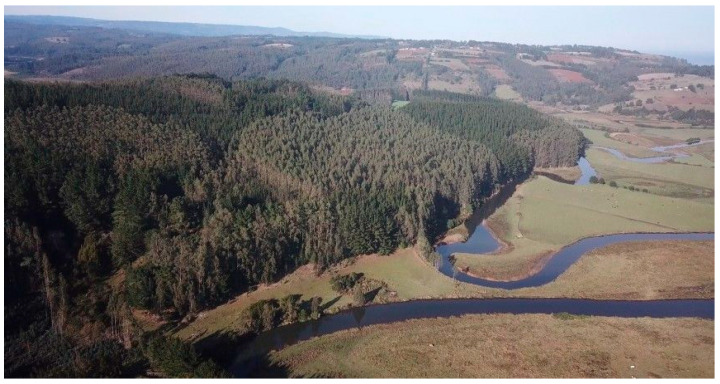
Monocultures of non-native trees in Mapuche territories (September 2019) [62].

**Figure 4 ijerph-20-05672-f004:**
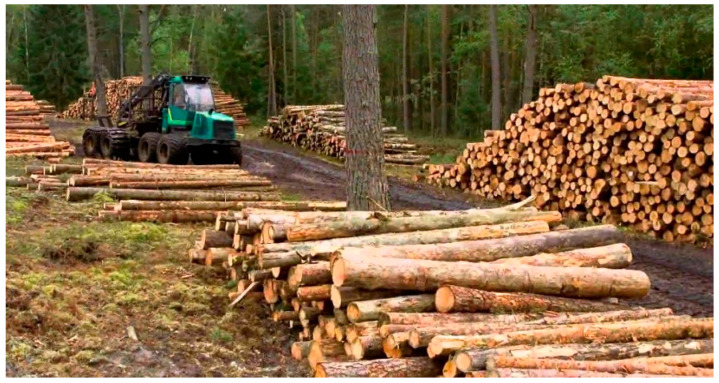
Forest extractivism in Mapuche territories (April 2021) [63].

**Table 1 ijerph-20-05672-t001:** Summary of the sample.

Unit of Analysis	Individual
Sampling	54 out of 67 communities
Number of participants	46 kimeltuchefes
Gender	Female and male
Key participants	Kimeltuchefes: Logko, Kimche Lawentuchefe, Norche, Ngenpin

**Table 2 ijerph-20-05672-t002:** Inclusion criteria for participants.

Criteria	Description
Key participants	Kimeltuchefes: Logko, Kimche Lawentuchefe, Norche, Ngenpin
Age	People over 60 years old
Gender	Female and male
Ability	Have the Mapuche kimün
Indigenous people	Mapuche

## Data Availability

Not applicable.

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
