# Peer review of "Implications of Extractivism and Environmental Pollution in Mapuche Territories of the Araucania Region"

_ijerph, 2023, doi:10.3390/ijerph20095672_

Round 1

Reviewer 1 Report (New Reviewer)

The authors present an interesting research on implications of extractivism and environmental pollution in mapuche territories of the araucania region.

Some suggestions to improve the document.

1-   In the abstract line 17 – the objective was …?!! This must be rephrase to the objective of this research is …..

2-   In the abstract What is Ngenh (line 24) ?

3-   In the abstract - What is this ?? - .az mapu, and ixofil mogen

4-   In the abstract what do the authors concretely mean with extractivism??

5-   In the abstract the use of not known acronyms it is not desirable, because it does not capture reader’s attention.

6-   Summary on the Abstract: The abstract is not capturing the “normal” reader’s attention… It musts be totally reformulated. Some of the elements from an Abstract are already there. Other are still missing.

7-   The authors should create a table with all the specific names from the Chile-indigenous people and the respective explanation of their meaning.

8-   The figures 2,3, and 4 should be dated. The date of shooting (month and year) should be provided in the description of the figures.

9-   The conclusions should be segmented. Probably the use of a table would help to get the most important conclusion in a snapshot.

10-                   In the conclusions is missing the academic implications and managerial implications.

Good luck

Author Response

Dear Reviewer, we listened to your suggestions, they are incorporated in the text and colored with light blue.

Reviewer 2 Report (New Reviewer)

Are there data about medicinal herbs lost by extractivism and environmental pollution at mapuche territory along the years? If there are these data, cite in results section.

Are there data about environmental pollution at mapuche territory along the years? If there are these data, cite in results section.

Is there pesticide analysis on water at Mapuche territory?

These data can support the perception of Mapuche people long the years. 

####---------

You conclude the environmental pollution cause same disease on Mapuche people lines 542 - 543. However, didn't show any data on results that support this affirmation.

Figure 1.:

- It is not clear to me whether the conceptual map shown in figure 1 is a result of the interviews or concepted by the authors. Please make this more clear. 

 - What are the numbers inside the parentheses for each subcategory?

Author Response

Dear Reviewer, we listened to your suggestions, they are incorporated in the text and colored with light blue.

Round 2

Reviewer 1 Report (New Reviewer)

good job

Reviewer 2 Report (New Reviewer)

ok.

This manuscript is a resubmission of an earlier submission. The following is a list of the peer review reports and author responses from that submission.

Round 1

Reviewer 1 Report

This manuscript reports a qualitative research to assess implications of extractivism and environmental pollution in Mapuche territories of the Araucania region (Chile). I think that the research should be of a certain interest but in my opinion it does not fit well with the aim of the journal, sepcifically wi the aim of the special issue Environmental Exposure, Health effects and Risk. In fact, no data on none of this topics is reported in the manuscript. Anyway, the main flaws of the manuscript concern the methods. The authors performed interviews and integrated participant observation to unveil the implications of extractivism and environmental pollution in Mapuche territories in the Araucanía region. They selected  46 participants from Kimeltuchefe people located in rural Mapuche communities, that were grouped in 5 categories depending on their knowledge, wisdom, and experience. Firts, the number of persons selected is too low, mainly considering the 5 groups. Second, it is not clear how the authors selected the persons. Third, it was never stated what kind of questions were addressed to participants. Overall, the results and the conclusions of the study are quite obvious.

Author Response

Dear Reviewer, along with greetings and hoping you are well, through this we respond to each of your observations. Warm greetings.

Response to reviewer 1.

Reviewer Comments 1

Response to comments

Point 1.

I believe that the research should be of some interest but in my opinion it does not fit well with the objective of the journal, specifically with the objective of the special issue Environmental Exposure, Health Effects and Risk. In fact, no data on any of these topics is reported in the manuscript.

Regarding point 1, we respectfully point out that:

The objective of the study was: to unveil the implications of extractivism and environmental pollution in Mapuche territories in the Araucanía region.

In attention to the objective, it is revealed that "monocultures of non-native trees", "environmental pollution" and "forest extractivism". They violate the sacred spaces of Mapuche territories. Since it generates soil degradation, water pollution and scarcity of these. This negatively disturbs the existence of biodiversity. In addition, it has negative implications for the agricultural activities carried out by the Mapuche. All this has negative effects on the health of the Mapuche, and therefore puts the health of these individuals at risk, as well as the subsistence of their communities.

In addition, "monocultures of non-native trees", "environmental pollution" and "forest extractivism" disturb and break the foundamentals under which the Mapuche govern their way of life. Among these foundamentals are: az mapu; kume mogen; Ixofil mogen; reciprocity, which are closely linked to the way of understanding and approaching health from a comprehensive perspective. Since the health of the Mapuche is linked to nature. In turn, these foundamentals support the "Mapuche worldview", which alludes to a way of perceiving and understanding their environment, and the world, reflected in every aspect of daily life (Rodríguez and Saavedra, 2011).

Consequently, "monocultures of non-native trees", "environmental pollution" and "forest extractivism" cause risks, and therefore, negative effects on the health and daily life of the Mapuche. This is evidenced in the: background of the introduction, results, discussion and conclusions. Therefore, the article fits with the objective of the special issue of the magazine.

Finally, add that more evidence was incorporated regarding "health effects and risk", in the:

1. Introduction.   

Point 2.

46 participants from the town of Kimeltuchefe located in rural Mapuche communities were selected, who were grouped into 5 categories according to their knowledge, wisdom and experience

Regarding point 2, we respectfully indicate that:

First of all, the Kimeltuchefe were not grouped into 5 categories as you indicate.

It should be noted that the category "forest extractivism and environmental pollution", which is related to the subcategories: a) Non-native tree monocultures; b) Environmental contamination; c) Forest extractivism; d) Az mapu; e) Küme mogen. These subcategories emerge from the narrations of the kimeltuchefes. In addition, to indicate that these subcategories achieved a high theoretical saturation

Point 3

First of all, the number of people selected is too low, considering mainly the 5 groups.

Regarding the 46 participants (kimeltuchefes) selected in the study, we respectfully point out that:

It is due to the special characteristics that these participants have, that is, those who have the category of kimeltuchefe were selected, and therefore, those who have a better knowledge of the phenomenon investigated. In addition, these participants are located in the place where this phenomenon occurs (Forest extractivism and environmental pollution). The above is explained in 3.1. participants

Also note that people who have the category of kimeltuchefes are scarce. Since they possess the Mapuche kimün (It alludes to Mapuche knowledge, in the different fields of human development: social, cultural, economic, education, among others). The kimeltuchefes are elderly people. The following arguments are then displayed:

The kimeltuchefes are people who have acquired the ability and methodology to teach during their lives. They are able to deliver the Mapuche kimün (Pereira, Reyes and Pérez, 2014). Likewise, Augusta (1903) points out that in the case of the Mapuche culture, the holders of knowledge are the grandparents who have the category of kimeltuchefes.

In this context, it is also worth pointing out that kimeltuchefes are currently scarce, given that the new generations have been losing the Mapuche kimün, due to the imposition of Eurocentric culture by the Chilean State

Also add that the selection of participants (kimeltuchefes) is consistent with qualitative research, since its objective is to capture reality in the context in which the phenomena occur, and therefore, allows understanding these phenomena. In this sense, qualitative research allowed us to capture reality and understand the phenomenon, based on the perception that the kimeltuchefes have of their own context.

All of the above is explained and argued in the header:

3. Materials and Methods

3.1. Participants

Finally, note that qualitative research does not generalize the results; it is not focused on studying large numbers of participants, but studies subjects in a specific context; and for the rest, it is not centered in the numerical.

Point 4.

Second, it is not clear how the authors selected the people.

In point 4, we indicate that:

The selection of the participants was made intentionally. This is explained and argued in the header:

3. Materials and Methods

3.1. Participants

Likewise, his suggestion was considered, further detailing the way in which the participants (kimeltuchefes) were selected. The above is explained in the header:

3. Materials and Methods

3.1. Participants

Point 5.

Third place, it was never indicated what type of questions were addressed to the participants

Regarding point 5:

It should be noted that the type of questions addressed to the participants (kimeltuchefes) is supported by Mella and Osses (2015). The above is explained in the header:

3. Materials and Methods

3.2 Instruments

Likewise, for greater clarity, your suggestion was considered, regarding the type of questions addressed to the participants (kimeltuchefes). The above was incorporated into the header:

3. Materials and Methods

3.2 Instruments

Point 6.

In general, the results and conclusions of the study are quite obvious

Regarding the comment in point 6, we respectfully indicate that:

The results and conclusions are not quite obvious as you indicate:

First, because there are no investigations that have addressed the objective set out in this study, and even more so in the Araucanía region, from the qualitative methodology.

Second, there is no evidence of other studies that have addressed the objective set forth in this study, from the narrations of the kimeltuchefes, and from participant observation.

Third, "monocultures of non-native trees", "environmental pollution" and "forest extractivism" have negative effects on the loss of water and biodiversity. As well as, they transgress and negatively affect the: az mapu; kume mogen; ixofil mogen; reciprocity, which are very important foundations in the Mapuche worldview, since they govern the life of the Mapuche.

In view of the above, the results and conclusions show negative effects and risks on health, and on the subsistence of the Mapuche. Therefore, an imbalance between the Mapuche and nature is visualized.

In short, from the perspective in which the research was carried out, novel results and conclusions are displayed.

The aforementioned, is observed in depth, in the headers: results, discussion and conclusions.

Reviewer 2 Report

The research conducted by Beltrán-Véliz et al. is scientifically interesting, the authors made an effort to assess the implications of extractivisms and environmental pollution for Mapuche community in Chile, moreover they adress the consequences of natural resources overexploitations for rural inhabitants also on their every-day life and their chances to live an healthy life also for the next generation in combination with the health of the ecosystem. Therefore, the paper perfectly match with the scope of the journal and it is ready for publication. I suggest to put some more pictures, if they have any, after each subcategory in the Results paragraph.  

Author Response

Dear Reviewer, along with greetings and hoping you are well, through this we respond to each of your observations. Warm greetings.

Response to reviewer 2

Reviewer Comments 2

 Response to comments

Point 1.

A minor spell checker is required.

Regarding point 1, we point out that:

The suggestion was considered

Point 2.

I suggest incorporating more photos, yes there are

Regarding point 2, more photographs were incorporated

Round 2

Reviewer 1 Report

I thank the authors for they reply to my concerns. However, I reiterate my comments from the first revision round. This is a qualitative research performed on a limited sample and methods of the survey should be better explained.

Author Response

Dear Editor, along with greetings, we have accepted most of your suggestions, which you can see in the manuscript (highlighted in yellow).

Warm greetings.
